# Comparative in Mechanical Behavior of 6061 Aluminum Alloy Welded by Pulsed GMAW with Different Filler Metals and Heat Treatments

**DOI:** 10.3390/ma12244157

**Published:** 2019-12-11

**Authors:** Isidro Guzmán, Everardo Granda, Jorge Acevedo, Antonia Martínez, Yuliana Dávila, Rubén Velázquez

**Affiliations:** 1Faculty of Systems, Universidad Autónoma de Coahuila, Saltillo 25280, Mexico; yuliana_avila@uadec.edu.mx (Y.D.); rvelazquezmora@uadec.edu.mx (R.V.); 2UAEM University Center at Atlacomulco, Universidad Autónoma del Estado de México, Toluca 50000, Mexico; 3Corporación Mexicana de Investigación en Materiales, Saltillo, 25290, Mexico; jacevedo@comimsa.com; 4Faculty of Chemical Sciences, Universidad Autónoma de Coahuila, Saltillo 25280, Mexico; aml15902@uadec.edu.mx

**Keywords:** precipitation hardening aluminum alloys, weld joint, GMAW, tensile strength, solubilization heat treatment, mechanical properties

## Abstract

Precipitation hardening aluminum alloys are used in many industries due to their excellent mechanical properties, including good weldability. During a welding process, the tensile strength of the joint is critical to appropriately exploit the original properties of the material. The welding processes are still under study, and gas metal arc welding (GMAW) in pulsed metal-transfer configuration is one of the best choices to join these alloys. In this study, the welding of 6061 aluminum alloy by pulsed GMAW was performed under two heat treatment conditions and by using two filler metals, namely: ER 4043 (AlSi_5_) and ER 4553 (AlMg_5_Cr). A solubilization heat treatment T4 was used to dissolve the precipitates of β”- phase into the aluminum matrix from the original T6 heat treatment, leading in the formation of β-phase precipitates instead, which contributes to higher mechanical resistance. As a result, the T4 heat treatment improves the quality of the weld joint and increases the tensile strength in comparison to the T6 condition. The filler metal also plays an important role, and our results indicate that the use of ER 4043 produces stronger joints than ER 4553, but only under specific processing conditions, which include a moderate heat net flux. The latter is explained because Mg, Si and Cu are reported as precursors of the production of β”- phase due to heat input from the welding process and the redistribution of both: β” and β precipitates, causes a ductile intergranular fracture near the heat affected zone of the weld joint.

## 1. Introduction

Precipitation hardening aluminum alloys of the 6xxx series are extensively used in many industries because their good corrosion performance, machinability, weldability, as well as their relatively low cost [1]. Structural alloys of aluminum type 6061 are used in the cooling systems of nuclear reactors due to their high transparency to neutrons, low radiation-induced heating and excellent thermal conductivity [2]. It is also used in the automotive industry in high-end cars because of its lightness, which favors the reduction in the emission of greenhouse gases [3,4].

The hardening β-phase containing Al, Mg and Si, is encountered in this type of aluminum alloys, whose precipitation sequence is as follows: super saturated solid solution (SSSS) → Si/Mg cluster → Guinier–Preston (GP) zones → β” → β’ → β, where β provides the best mechanical resistance to the alloy [5,6,7].

The excellent weldability of the 6061-T6 by the metal inert gas (MIG) or gas metal arc welding (GMAW) techniques has been demonstrated, and some studies intended to determine the weldability and the best mechanical and microstructural properties of this material have been developed. Chikhale et al. [8] studied the microstructural characteristics of a 6061-T6 alloy welded by the GMAW process; they used several welding parameters and an ER 4043 electrode. A maximum tensile strength of 176 MPa was achieved when a current of 175 A and 23 V in electric potential difference were applied to the electrode. Also, Vickers hardness values averaging 62 to 70 HV were reported in the weld zone (WZ), the heat affected zone (HAZ) and the base metal (BM). 

Patel et al. [9] studied the GMAW process variables adjusting the torch angle, current and voltage. The best weld penetration values were 5.3 mm (22 V, 100 A) and 6.3 mm (24 V, 120 A) achieved at 50° torch angle. Recrystallized grains were observed near the fusion zone (FZ) while elongated grains were noted in the BM, attributed to the heat conduction during the welding process.

Missori and Sili [10] evaluated the mechanical and microstructural behavior of 6082-T6 alloys welded by GMAW. They found that by applying 200 A and a feed rate of 0.387 mm/min, the maximum tensile strength reached 180 MPa and the hardness value near the cross-section of the WZ was 95 HV. As a relevant remark, they reported that the fracture occurred in the HAZ; in agreement with other authors that found similar findings, where the fracture mechanism was established as a ductile intergranular failure with evidence of some fragile zones [11].

In order to determine the elements which promotes the formation and distribution of precipitates during GMAW and the consequent affectation of mechanical properties of 6xxx alloys, several characterization techniques have been used. The hardening β”-phase is dissolved due to the heat input from welding procedure and it has been identified after GMAW by using scanning electron microscopy (SEM), differential scanning calorimetry (DSC) or Transmission electron microscopy (TEM) [12,13]. During GMAW, the HAZ is susceptible to the formation of β” and β precipitates grouped as needles and sticks because of high temperature (256–266 °C), leading to the formation of fragilization zones. Mathematical simulation demonstrated similar results, as it was reported by [14]. Moreover, Mg, Si and Cu have been identified as the precursors of the precipitation of β”, whose particle size grow with time and ageing temperature. The latter has been found by atom probe tomography (APT), as well as the phase field crystal (PFC) technique [15].

In this investigation, the comparative analysis of the GMAW process applied to a 6061 aluminum alloy by using two filler metals: ER 4043 and ER 5356, as well as two different thermal treatments T6 (as received) and T4 (solubilization heat treatment) is presented. The aim of this analysis is to find the best welding conditions and to explain the effect of the thermal treatment on the weldability and tensile strength properties of the weld joints. Also, it could help to explain the mechanism of failure of the samples due to the re distribution of β” and β precipitates in the vicinity of the HAZ due to the heat flux from the welding process. The solubilization heat treatment, together with the use of an adequate filler metal and proper welding parameters leads to the production of good quality weld joints and increase their tensile strength.

## 2. Materials and Methods 

For this investigation, 32 pieces of 6061 aluminum alloy with dimensions 150 mm × 100 mm and a thickness of 7 mm were prepared. A V-groove cut with 45° bezel was realized in the border of each piece. 16 pieces were kept in the original heat treatment of the alloy (as received, T6 heat treatment) which is an artificial ageing treatment consisting in the solubilization at 540 °C for stress relieved during 6 h followed by artificial ageing at 155 °C during 5 to 18 h; finally the material is quenched in water at 100 °C. Meanwhile, other 16 pieces were put in a Thermolyne muffle furnace, (Thermo fisher Scientific company, Waltham, MA, USA) during a soak time of 3 h at 350 °C (standard solubilization treatment T4) to dissolve the β”-phase [16].

Two types of electrodes were used for welding: AlSi_5_ (ER 4043) and AlMg_5_Cr (ER 5356). Spark optical emission spectroscopy (Ametec materials division, Kleve, Germany) was used to determine the chemical composition of the aluminum alloy base metal (AA 6061) and both filler metals: ER 4043 and ER 5356, as it is observed in Table 1.

Two pieces with the same heat treatment condition were put together with the V-groove facing each other and the resulting joint is considered as a weld sample. The welding equipment was a Lincoln Power Wave S500, (Lincoln Electric, Clev, OH, USA) able to perform GMAW pulsed incorporating an argon atmosphere to protect from the oxidation. The bias voltage was set to 20 V and the current was adjusted from 250 to 295 A in the set of experiments showed in Table 2, where HT is the heat treatment, I is the welding current, $w_{s}$ is the welding speed, $Q_{net}$ is the heat input and $Q_{arc}$ is the arc energy, calculated as follows [17,18]: (1)$$Q_{net}=\eta \cdot Q_{arc}$$
(2)$$Q_{arc}=\frac{E\cdot I}{w_{s}}$$
where E is the bias voltage and η the heat transfer efficiency, commonly defined as 0.86 for GMAW processes [19].

The selection of processing parameters was performed in accordance to the literature review, where the main variables are: (a) heat treatment condition [20], (b) voltage and current during the welding process [21] and (c) welding speed [22].

After the welding procedure, the samples were prepared to metallographic examination, as it is stablished in the ASTM E3 standard [23], upon the following steps: a) cross-sectioning cut on the weld joint zone and sample mounting in Bakelite, b) surface of weld samples gradually ground with SiC paper, and c) polishing to a mirror-like finishing with 1 μm diamond paste and SiO_2_. A mean roughness (Ra) in the range from 0.04 to 0.05 µm was determined for the surface of the welded samples by using a Surtronic S125 roughness meter (Taylor Hobson, Berwyn, PA, USA). Finally, samples were chemically etched with a 3% hydrofluoric acid solution, in accordance to the ASTM E407 standard [24], to reveal the chemical phases and to observe the HAZ.

An optical microscope Nikon Eclipse MA200 was used to observe metallographic samples and the acquired images were processed by NIS Elements DS-03 software (Nikon, Brighton, MI, USA) to analyze the microstructure of base metal, weld zone and HAZ. The size of precipitates and the proportion of constituent phases were measured also. Afterwards, the weld quality parameters, namely: welding crown and weld penetration depth, were measured in a Nikon stereoscope, meanwhile the average equivalent area of pores was obtained by means a Mira2 X-Sense SEM instrument (Tescan, Libusina Trida, Brno, Czech Repubic). Energy Dispersive X-ray Spectroscopy (EDS) model, was used to identify chemical elements.

The phase identification was performed in a Empyrean X-ray diffractometer (XRD, (Malvern Panalytical, Etten Leur, The Netherlands) in micro diffraction mode on a 1 mm × 1mm surface. Phase analysis was realized in the Highscore Plus Software 4.8, (Malvern Panalytical, Eindhoven, The Netherlands) and the Powder Diffraction File Plus (PDF+) database. The settings for the XRD test were set to 0.0016° in step size, 87.92 s in count time, CuK_α_ radiation, accelerating potential of 45 kV and 40 mA in electron current. Rietveld refinement method was used in a 2θ range from 35° to 42°. The latter is a theoretical method to adjust an XRD pattern by a model where experimental and structural variables are included [25,26].

A Rockwell 2000 durometer (Instron, Norwood, MA, USA)) was used to measure the hardness of base metal (Rockwell 15T). The Vickers microhardness profile in the welding region was evaluated by performing 40 micro indentations along each sample by using an Instron Tukon 2500 tester (0.3 kgf load) [27].

Samples were cross-sectioned and machined as plates of 12.5 mm in width, 200 mm in overall length, 6.35 mm in thickness, 57 mm in reduced section and 50 mm in gage length, as preparation to stress tests in accordance to the ASTM E8 standard [28]. The strength test is performed from averaging the results of two test specimens (a sample and a replica), upon recommendation of the AWS D1.2/D1.2M standard [29]. Then, two test specimens from each pair of welded plates were obtained and submitted to the stress test in an Instron 5980 tensiometer. The test was performed at a yield speed of 8 MPa/s and a stress speed of 10 mm/min, to determine the ultimate tensile strength (UTS), Young’s modulus, yield stress and final elongation.

## 3. Results

### 3.1. Microstructure and Phase Identification after Heat Treatment

After the solubilization heat treatment T4, the microstructure of the aluminum alloy was compared against the original condition of the material (T6 heat treatment). Figure 1a exhibits the microstructure of the material as received, exhibiting acicular precipitates dispersed in the aluminum matrix, which correspond with β”-phase (Mg_5_Si_6_). In contrast, the micrograph of the Figure 1b shows precipitates of β-phase (Mg_2_Si), formed after the solubilization heat treatment as strip and rods precipitates. Because of the heat treatment, the particle size of precipitates increased from a mean of 3.85 µm, standard deviation 2.29 µm (material as received) to 4.12 µm, standard deviation 2.73 µm (T4 heat treated material). Also, the Rockwell hardness of the original material, measured in 85 HRT-15, was reduced to 60 HRT-15 after the solubilization heat treatment.

The presence of Mg, Si and Cu, as main precursor elements of the β”-phase was confirmed by EDS analysis, as it can be observed in Figure 2 for both the original material and the alloy after solubilization heat treatment. The latter has been established in the literature by using atomic resolution high-angle annular dark field scanning transmission electron microscopy and atom probe tomography [30]. The acicular precipitates have been reported as indicators of the presence of a greater content of Mg than Si with ratios of Mg:Si of 1.21:1, in contrast with the widened and planar precipitates, leading in a reduction of the Mg content to a ratio of 1.23:1 [1].

The XRD diffractograms of both: untreated T6 and heat treated T4 samples are exhibited in Figure 3. The untreated sample (as received) has a T6 heat treatment, which consists in solubilized at 540 °C during 6 h and artificial ageing at 155 °C during 5 h [16]. As a consequence, characteristic reflections of β”-phase belonging to Mg_5_Si_6_ were identified, as it was reported by Zandenberg [31]. Lattice parameters a = 15.9 Å, b = 4.05 Å, c = 6.74 Å and β=105.3° were calculated, as well as a C_2_ monoclinic space group, usually identified by a needle-like microstructure. In the case of the T4 heat-treated sample, its XRD spectrum is exhibited in Figure 4. This sample showed the presence of plates of β-phase (Mg_2_Si) of cubic space group and a = 6.34 Å in its microstructure. Also, the recrystallization of the aluminum matrix is notorious.

### 3.2. Tensile Strenght Evaluation

The results of the tensile strength test are showed in Table 3 for each weld sample, reporting the UTS, rupture load, Young’s modulus, yield stress and final elongation. Best results in UTS value were obtained for samples M8-T6-4043 and M2-T4-4043, reaching up to 153 MPa followed by the sample M4-T4-5356 (UTS = 113 MPa) and M7-T6-5356 (UTS = 101 MPa). The tensile strength evaluation allowed to select representative samples to explain the mechanisms involved in their failure; then it could be used to improve and understanding of the welding process.

The tensile strength as a function of the welding current is exhibited in Figure 5, as the most relevant mechanical parameter to be evaluated after welding. Results indicated that the ER 5356 filler metal has a more stable behavior than the ER 4043 for both thermal treatment conditions. The best results for tension tests when ER 5356 was used were encountered at higher welding current values, without a notorious effect of the heat treatment. In contrast, when the ER 4043 filler metal was used, the solubilization heat treatment T4 slightly improves the tensile strength at intermediate values of welding current, but the effect at low or high current is negligible.

### 3.3. Characterization of the Weld Joints

After processing by pulsed GMAW, each weld sample was characterized to evaluate different measures of weld quality, and they are shown in Table 4. In accordance with the AWS D1.2 standard [29], the minimum weld crown must be greater than 0.25 mm, the maximum average porosity should not exceed 3 mm and it is recommended full weld penetration (>7 mm in this experimentation). With exception of samples M5-T6-4043, M6-T6-4043 and M7-T6-4043, all samples exhibited good quality indicators.

Micrographs for the samples that reported the best tensile strength performance were selected and are exhibited in Figure 6. The weld zones were indicated: HAZ, FZ and WZ; also, the weld imperfections were marked. The best samples welded with ER 4356 were M4-T4-5356 (113.9 MPa) for T4 heat treatment, showed in Figure 6a, and M7-T6-5356 (101.9 MPa) for T6 heat treatment, exhibited in Figure 6b. On the other hand, best samples welded with ER 4043 were M2-T4-4043 (153.9 MPa) for T4, shown in Figure 6c, and M8-T6-4043 (153.6 MPa) for T6, showed in Figure 6d. Selected samples exhibited full weld penetration, complete fusion, adequate weld crown and low porosity.

The grain size in the region surrounding the fusion zone was measured in the same selected samples, to analyze the effect of thermal treatment and welding process on this parameter. Sample M7-T6-5356, with the original T6 heat treatment, exhibited a mean grain size of 41.4 µm (standard deviation, SD = 11.27 µm) while the sample M4-T4-5356, submitted to the solubilization heat treatment T4, reported an average of 53.4 µm (SD = 15.08 µm). Similar results were encountered for the samples welded with ER 4043 filler metal, where the sample M8-T6-4043 showed a mean grain size of 42.1 µm (SD = 13.83 µm) and the sample M2-T4-4043 exhibited an average of 59.57 µm (SD = 8.84 µm). 

In order to evaluate the hardness in the different regions of the weld joint, a cross-section hardness profile was performed. Hardness of the weld region varies between 50 and 80 HV, as it can be observed in Figure 7, but each region is clearly identified by its characteristic hardness regime. The hardness of the HAZ reached the highest values, while the Weld Zone presented the lower ones. Sample M2-T4-4043 shows an increase in hardness in the HAZ and a low-hardness ductile region is present in the FZ. Finally, it is remarkable that the samples welded with ER 4043 presented higher hardness values than the samples welded with ER 5356; even, samples welded with this filler metal exhibited homogeneous hardness along the entire weld joint.

Figure 8a presents the XRD spectrum from the HAZ of sample M8-T6-5356 where a 16.3% of β-phase and 83.7% of Al were estimated. In contrast, Figure 8b shows the XRD spectrum for the sample M2-T4-4043, specifically in the HAZ; the presence of 48% of β-phase (Mg_2_Si) was determined, while 52% corresponds to Al.

### 3.4. Fracture Behavior

The evaluation of micrographs from the samples after the rupture test were realized by SEM. All samples presented the fracture in the FZ without any exception and micrographs of selected samples are exhibited in Figure 9. The failure mechanism corresponds to a ductile fracture, associated with the nucleation and growing of microcavities and coalescence of microholes [13,32]. It is noticeable that the nucleation centers in the cavity were promoted by the presence of nanoparticles whose morphology and size were heterogeneous. Also, the average porosity was estimated in 8.74 mm.

## 4. Discussion

The solubilization heat treatment T4 promoted the change in the morphology of aluminum matrix from dispersed acicular precipitates (typical of β”-phase) to a morphology of strips and rods precipitates (typical of β-phase). The reduction in hardness after the solubilization heat treatment and the change in morphology of the precipitates is also an indicative of the thermal dissolution of β”-phase clusters to β-phase [32,33]. In the literature, the solubilization thermal treatment has been performed to 6061-T6 aluminum alloy followed by natural ageing during up to 24 h: the dissolution of β” precipitates, the diminishing oh hardness from 120 HV to 90 HV and the change in precipitates morphology from acicular to rods were reported [20]. This is attributed to the thermal effect on some alloy elements, such as Mg, Si and Cu, which reallocate in the aluminum matrix leading to the formation of new phases [4]. The evidence of such a phenomenon is confirmed by EDS and XRD analysis.

The tensile strength of the weld samples is also affected by the dissolution of β”-phase in the HAZ and the region adjacent to FZ, from a SSSS → Si/Mg cluster → GP zones → β” → β’ → β [34,35,36,37]. The sequence of phase transformation in 6xxx alloys is also denoted by the change in morphology and size of the grains in the aluminum matrix because of changes in temperature [38]. The phenomenon of phase transformation is related to input heat, which in turn is strongly dependent from both: weld speed and welding current [39]. In this investigation, M2-T4-4043 was the sample with the best performance in tensile strength reaching up to 153 MPa in maximum stress while the worst case was the sample M7-T6-4043 with 64 MPa. The first sample was welded by applying 270 A, at 0.39 cm/s in weld speed, whose estimated net heat input was established in 1.2321 kJ/mm; in contrast, the second sample was welded with 280 A, 0.61 cm/s resulting in a net heat input of 0.819 kJ/mm.

Weld penetration is directly proportional to the arc energy in a GMAW process, as it was studied by Chen [34]. However, our results indicate that arc energy also directly influences the hardness of weld joint and this effect is more acute in the fusion zone. The hardness behavior is very similar to other welding process of aluminum, such as MIG [35], where the hardness variations across the different welding regions have been reported, and they are attributed to the precipitate dissolution in the HAZ and FZ [8,9,10]. Grains in the HAZ grow up in the BM direction causing variations in hardness from 50 to 80 HV; however, this is not the main promotor of thermal softening but the variation in the precipitation of β”-phase along the several weld regions [35].

The increase in grain size in HAZ and FZ as a consequence of higher net heat input is an evidence of the phase transformation mechanism to form β, which provides the tensile resistance to the material [8,40,41]. However, there is a compromise with the fracture mechanism, because the rupture near the FZ is attributed to the growth of equiaxial elongated grains because of the excessive heat flux from the welding process [36,42]. In fact, the occurrence of the fracture in the HAZ is the most common condition after welding 6xxx aluminum alloys [35].

The application of GMAW process generated hard zones in the weld joint, generally near the HAZ, but also softer regions in the FZ and WZ, due to the redistribution of the hardening phases. After welding, the micro-XRD analysis showed the presence of β-phase in the fracture zone, which promoted the occurrence of the fracture near the interface with the FZ. The failure mechanism was determined as intergranular ductile rupture this zone for all samples.

## 5. Conclusions

A comparative analysis of the GMAW pulsed process in joints of a 6061 aluminum alloy by using two filler metals (ER 4043 and ER 5356), as well as two different thermal treatments T6 (material as received) and T4 (solubilization heat treatment) was performed. The GMAW pulsed process demonstrates to be an effective method to weld precipitation hardening aluminum alloys if the adequate combination of welding parameters, filler metal and previous heat treatment are selected.

Our results exhibited that the ER 5356 filler metal (AlMg_5_Cr) produces weld joint with good quality indicators and a maximum tensile strength of 113.9 MPa in the sample with the original heat treatment condition T6 welded at 295 A. The solubilization heat treatment T4 produces negative effects in the sample welded with this filler metal at the same processing conditions (295 A, 20 V) achieving 94.4 MPa only. This is attributed to the increase in the grain size due to the combination of both: solubilization thermal treatment and grain recrystallization due to the excessive heat from welding process. However, the tensile strength of the sample in T4 condition welded at 280 A reached up to 110 MPa. Then, the proposed solubilization heat treatment promotes stronger weld joints at low current welding.

Samples welded with ER 4043 (AlSi_5_) exhibited the strongest weld joint, and the best result in tensile strength was achieved by the sample submitted to the T4 heat treatment reaching 153.9 MPa (270 A). Similar results (153.6 MPa) were obtained in the sample in the original T6 heat treatment, but it was welded by using 295 A. Again, the solubilization heat treatment increased the performance of the welding at lower current. The latter could avoid other adversarial effects, such as the excess of fusion, structural damage of the aluminum matrix and excessive heat affected zones. However, more experiments should be done to analyze this effect and to optimize the welding process.

In all tested samples, the fracture occurred near the fusion zone, where an intergranular ductile mechanism of failure was identified, because the heat flux from the welding process promotes the formation of zones with important changes in hardness attributed to the redistribution of precipitates and the recrystallization of the aluminum matrix.

## Figures and Tables

**Figure 1 materials-12-04157-f001:**
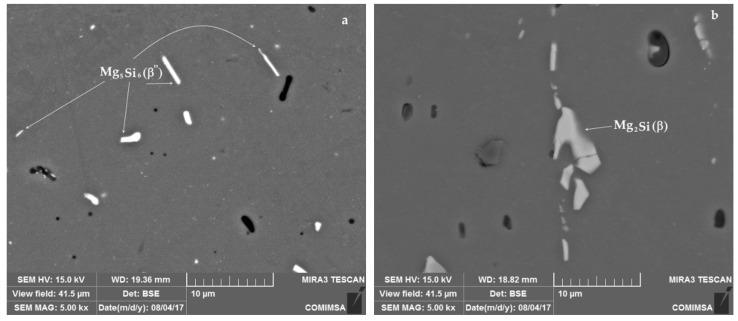
Microstructure of 6061 aluminum alloy (**a**) in the original T6 heat treat condition and (**b**) after the solubilization heat treatment T4.

**Figure 2 materials-12-04157-f002:**
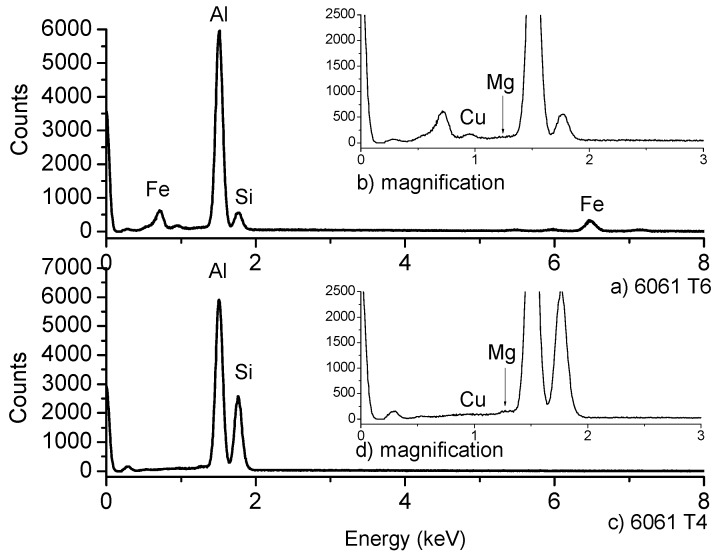
EDS analysis of the 6061 alloy, (**a**) in the original condition T6, (**b**) magnification to appreciate Mg and Cu content, (**c**) after solubilization heat treatment T4 and (**d**) magnification of EDS T6 sample.

**Figure 3 materials-12-04157-f003:**
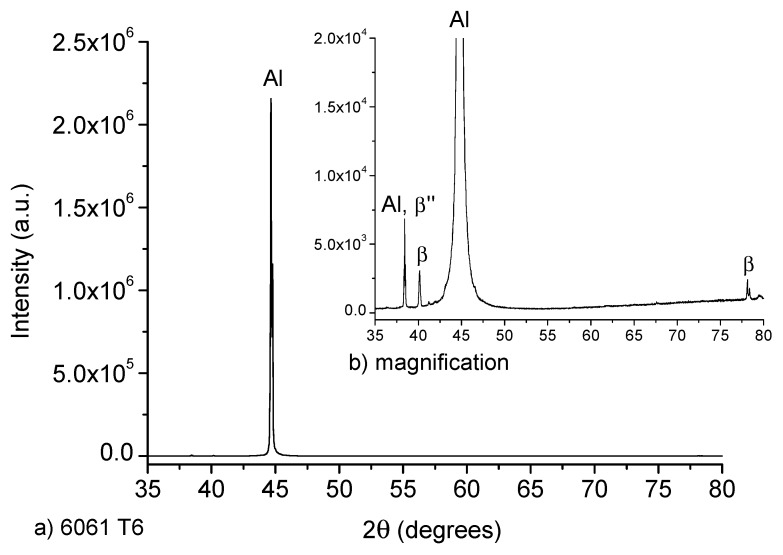
(**a**) XRD spectrum for 6061 alloy in the T6 heat treatment condition (as received), (**b**) magnification of the diffraction peaks corresponding to the hardening phases.

**Figure 4 materials-12-04157-f004:**
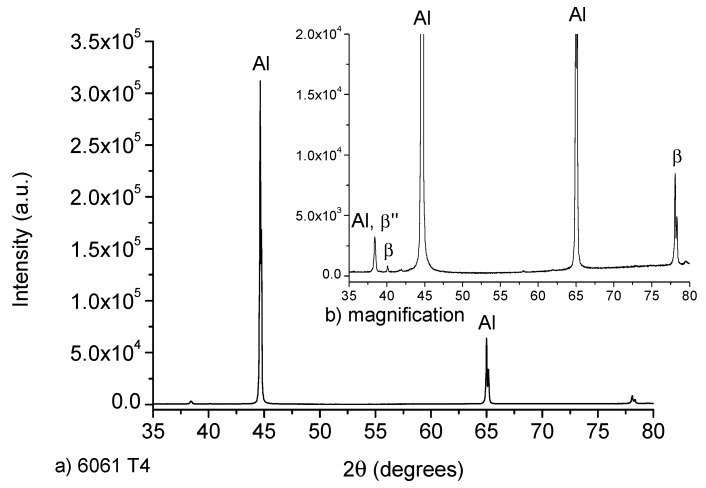
(**a**) XRD spectrum for 6061 alloy in the T4 heat treatment condition and (**b**) magnification of the diffraction peaks corresponding to the hardening phases.

**Figure 5 materials-12-04157-f005:**
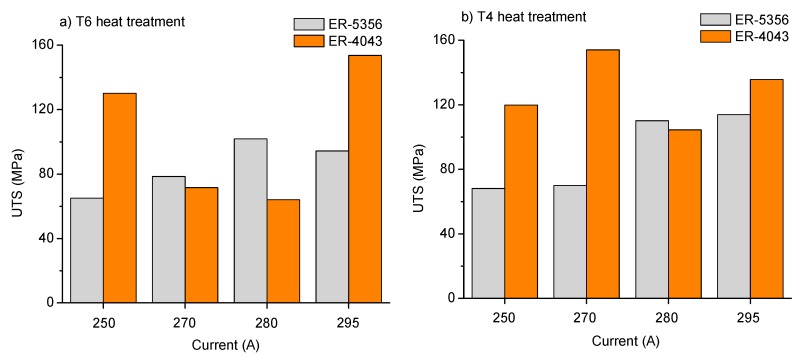
Behavior of tensile strength as a function of the welding current in (**a**) T6 and (**b**) T4 heat treatment conditions for the two tested filler metals.

**Figure 6 materials-12-04157-f006:**
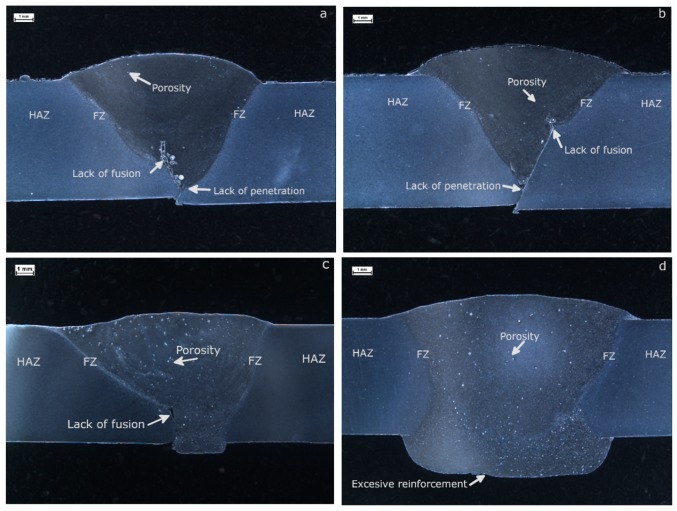
Microstructure of the weld joints exhibiting best performance in the tensile strength test with ER 5356 as filler metal (**a**) M4-T4-5356 and (**b**) M7-T6-5356, as well as ER 4043 (**c**) M2-T4-4043 and (**d**) M8-T6-4043.

**Figure 7 materials-12-04157-f007:**
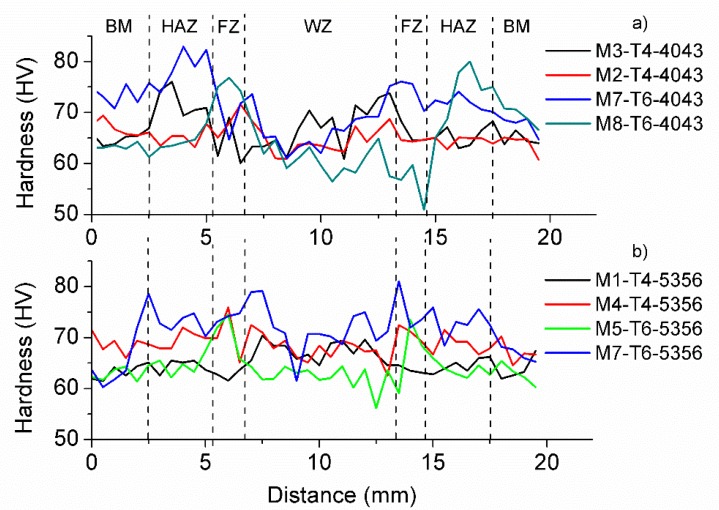
Cross-section hardness profile for selected samples indicating the variation across different weld regions (**a**) for samples welded with ER 40403 and (**b**) samples welded with ER 5356.

**Figure 8 materials-12-04157-f008:**
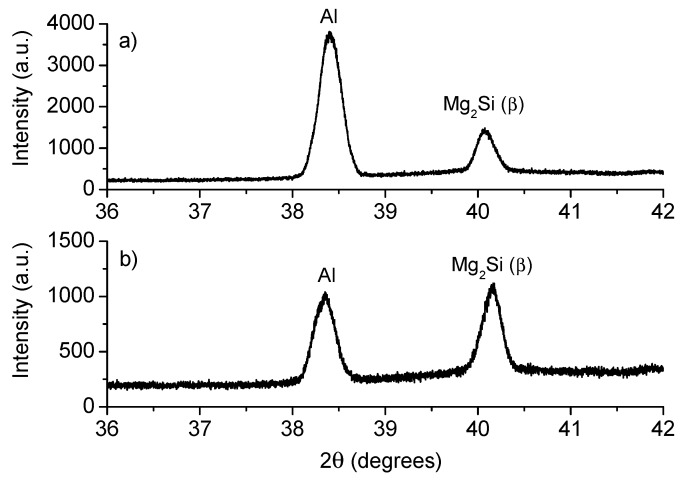
XRD spectrum from the HAZ of samples (**a**) M8-T6-5356 and (**b**) M2-T4-4043.

**Figure 9 materials-12-04157-f009:**
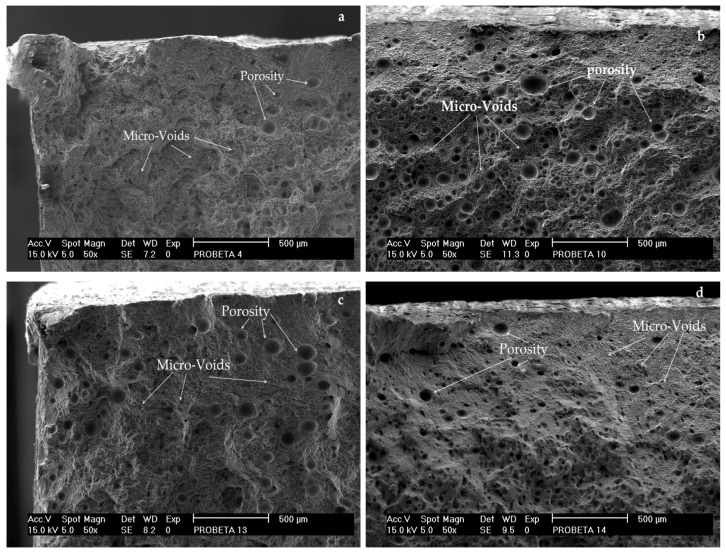
Fractographies of weld joints specimens: (**a**) M8-T6-4043, (**b**) M2-T4-4043, (**c**) M3-T4-5053 and (**d**) M7-T6-5053.

**Table 1 materials-12-04157-t001:** Chemical elements in the 6061 alloy and two filler metals.

Alloy	Composition in wt %							
	Si	Fe	Cu	Mn	Mg	Cr	Zn	Ti
AA 6061	0.80	0.70	0.40	0.15	1.20	0.35	0.25	0.15
ER 4043	6.00	0.80	0.30	0.05	0.05	-	0.10	0.20
ER 5356	0.25	0.40	0.10	0.10	4.60	0.70	0.10	0.11

**Table 2 materials-12-04157-t002:** Experimental parameters for GMAW pulsed of 6061 alloy.

Sample ID	HT	E (V)	I (*A*)	$w_{s}$ (mm/s)	$Q_{arc}$ (kJ/mm)	$Q_{net}$ (kJ/mm)
M1-T4-5356	T4	20	250	3.6171	1.3823	1.1888
M2-T4-5356	T4	20	270	2.9353	1.8397	1.5821
M3-T4-5356	T4	20	280	3.9357	1.4229	1.2237
M4-T4-5356	T4	20	295	3.7234	1.5846	1.3627
M5-T6-5356	T6	20	250	3.9978	1.2507	1.0756
M6-T6-5356	T6	20	270	3.8083	1.4180	1.2194
M7-T6-5356	T6	20	280	3.9347	1.4232	1.2240
M8-T6-5356	T6	20	295	3.8083	1.5493	1.3324
M1-T4-4043	T4	20	250	3.7310	1.3401	1.1525
M2-T4-4043	T4	20	270	3.8182	1.4143	1.2163
M3-T4-4043	T4	20	280	5.6000	1.0000	0.8600
M4-T4-4043	T4	20	295	3.9305	1.5011	1.2909
M5-T6-4043	T6	20	250	3.6567	1.3673	1.1759
M6-T6-4043	T6	20	270	4.0385	1.3371	1.1499
M7-T6-4043	T6	20	280	6.1000	0.9180	0.7895
M8-T6-4043	T6	20	295	3.8889	1.5171	1.3047

**Table 3 materials-12-04157-t003:** Summary of mechanical properties in tested samples.

Sample ID	UTS (MPa)	Rupture load (N)	Young’s Modulus (MPa)	Yield Stress (MPa)	Elongation (%)
M1-T4-5356	68.164	5449.32	44.260	44.2683	4.14
M2-T4-5356	70.000	3192.86	48.327	26.4598	3.40
M3-T4-5356 *	110.151	8874.82	105.090	54.8868	6.49
M4-T4-5356 *	113.900	9175.48	94.134	56.7094	9.08
M5-T6-5356	64.938	5183.30	93.378	27.9541	3.29
M6-T6-5356	78.548	6429.39	82.937	33.5481	4.31
M7-T6-5356	101.922	8219.11	94.904	46.5719	5.45
M8-T6-5356	94.402	7612.49	82.248	70.1278	5.44
M1-T4-4043	119.914	9503.50	94.775	70.5960	6.97
M2-T4-4043 ^#^	153.932	12334.26	93.590	64.4380	16.6
M3-T4-4043	104.543	7801.00	859.390	39.9000	39.00
M4-T4-4043	135.708	10806.14	80.804	54.1100	12.53
M5-T6-4043	130.140	10007.17	104.430	63.9230	3.79
M6-T6-4043	71.665	5759.22	80.484	66.9950	2.23
M7-T6-4043	64.000	4047.60	1315.650	35.8400	15.00
M8-T6-4043 ^#^	153.583	12404.39	76.913	74.6320	23.82

* Samples with highest UTS for ER 5356. ^#^ Samples with highest UTS for ER 4043.

**Table 4 materials-12-04157-t004:** Quality indicators of welding.

Sample ID	Weld Crown (mm)	Porosity Average Size (mm)	Weld Penetration (mm)
M1-T4-5356	1.6768	0.044125	8.3436
M2-T4-5356	3.0540	0.048770	10.3155
M3-T4-5356	1.2520	0.035675	8.5314
M4-T4-5356	1.7662	0.046635	8.3883
M5-T6-5356	0.5008	0.049290	11.250
M6-T6-5356	1.5739	0.045095	8.9115
M7-T6-5356	1.7886	0.067285	8.6475
M8-T6-5356	1.2207	0.033605	8.9115
M1-T4-4043	0.2638	0.045755	9.2268
M2-T4-4043	0.8658	0.043840	8.2528
M3-T4-4043	0.7674	0.007700	8.1362
M4-T4-4043	0.8500	0.060000	8.2536
M5-T6-4043	0	0.069310	9.1058
M6-T6-4043	0	0.049385	6.6030
M7-T6-4043	0.7087	0.010900	5.9249
M8-T6-4043	1.5516	0.046085	8.3436

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
