# Peer review of "Comparative in Mechanical Behavior of 6061 Aluminum Alloy Welded by Pulsed GMAW with Different Filler Metals and Heat Treatments"

_materials, 2019, doi:10.3390/ma12244157_

Round 1

Reviewer 1 Report

This manuscript mainly investigated the mechanical behavior of 6061 alloy welded by pulsed GMAW.

It is worth of publication after some compulsory revisions. The detailed comments and suggestions are as follows:

In Fig. 2, no peaks of Mg and Cu were indicated. Please check it.

In Fig. 5, the font size was too small. Please make it easier to read.

Reviewer 2 Report

This is an interesting work that fits within the scope of this Journal. To improve the quality and understanding of this article, the following points need to be addressed:

1. Abstract, page 1 line 20: replace ‘heat treat conditions’ with ‘heat treatment conditions’.

2. Abstract, page 1 line 20: replace ‘metal’ by ‘metals’. Also add the filler metals that you used.

3. Introduction, page 1 lines 37 and 39: replace ‘its’ by ‘their’.

4. Materials and methods: What was the resulting surface roughness after polishing?

5. Materials and methods: What was the concentration of the hydrofluoric acid that you used for etching?

6. Materials and methods: An analytical description of the heat treatments should be added in the experimental.

7. Materials and methods: The test conditions of mechanical tests should be also mentioned, instead of referring to AWS D1.2/D1.2M standard.

8. Results, page 3, paragraph 1: Please add experimental spread of mean values that present.

9. Consider using figures instead of tables, could help in showing any potential trend between examined parameters and test results.

10. The grain size (as you correctly mention in page 11, line 278) also plays a significant role in the mechanical performance of the alloy. Didi you measure the grain size for the different treatments?

11. Conclusions are missing.    

Round 2

Reviewer 2 Report

Thank you for addressing my comments/suggestions.